# Responsive Microgels through RAFT-HDA Dynamic Covalent Bonding Chemistry

**DOI:** 10.3390/molecules29061217

**Published:** 2024-03-08

**Authors:** Jingkai Nie, Hang Yin, Ruyue Cao, Changyuan Huang, Xiang Luo, Jun Ji

**Affiliations:** 1State Grid Smart Grid Research Institute Co., Ltd., Beijing 102211, China; niejingkai@geiri.sgcc.com.cn (J.N.); caoruyue@geiri.sgcc.com.cn (R.C.); jijun@geiri.sgcc.com.cn (J.J.); 2China Electric Power Research Institute Co., Ltd., Beijing 100192, China; huangchangyuan@epri.sgcc.com.cn (C.H.); luoxiang@epri.sgcc.com.cn (X.L.)

**Keywords:** ultrasound-responsive microgels, phosphoryl disulfide, furan groups, hetero Diels–Alder (HDA) reaction, dynamic covalent bonding

## Abstract

This paper developed a method for preparing ultrasound-responsive microgels based on reversible addition fragmentation chain transfer-hetero Diels–Alder (RAFT-HAD) dynamic covalent bonding. First, a styrene cross-linked network was successfully prepared by a Diels–Alder (DA) reaction between phosphoryl dithioester and furan using double-ended diethoxyphosphoryl dithiocarbonate (BDEPDF) for RAFT reagent-mediated styrene (St) polymerization, with a double-ended dienophile linker and copolymer of furfuryl methacrylate (FMA) and St as the dienophile. Subsequently, the microgel system was constructed by the HDA reaction between phosphoryl disulfide and furan groups using the copolymer of polyethylene glycol monomethyl ether acrylate (OEGMA) and FMA as the dienophore building block and hydrophilic segment and the polystyrene pro-dienophile linker as the cross-linker and hydrophobic segment. The number of furans in the dienophile chain and the length of the dienophile linker were regulated by RAFT polymerization to investigate the effects of the single-molecule chain functional group degree, furan/dithioester ratio, and hydrophobic cross-linker length on the microgel system. The prepared microgels can achieve the reversible transformation of materials under force responsiveness, and their preparation steps are simple and adaptive to various potential applications in biomedical materials and adaptive electrical materials.

## 1. Introduction

The Diels–Alder (DA) reaction [1] is a well-known and commonly used organic cycloaddition reaction in chemical synthesis, materials science, and biomedicine [2,3,4,5,6,7,8,9,10,11,12]. The central aspect of DA reactions is the combinational pairing of dienes and dienophiles, which gives birth to a wide variety of DA reactions. The retro-Diels–Alder reaction is a modern variation on traditional DA reactions [13]. The challenge with traditional DA reactions is their sluggish kinetics at room temperature when equilibrium is either reached slowly or at very high temperatures. To achieve quantitative conversion, research has been conducted on effective Diels–Alder reactions, such as reversible addition fragmentation chain transfer-hetero Diels–Alder (RAFT-HDA) reactions, to reach quantitative conversion within minutes at ambient temperature [14].

The research group of Sinwell, Barner-Kowollik, and others [15] proposed the RAFT-HDA reaction in 2008, based on the heteroatomic DA reaction between electron-absorbing dithioesters and cyclopentadiene or linear dienes. In general, an electron-absorbing dithioester acts as a chain transfer agent in reversible addition-break chain transfer (RAFT) polymerization to mediate monomer polymerization, followed by an HDA reaction with corresponding dienes in the form of a macromolecular chain end group as a dienophile. The RAFT-HDA reaction can be quantitatively converted in minutes to hours at room temperature or 50 °C, allowing for the preparation of block polymers in water without using a catalyst. The RAFT-HDA reaction benefits various applications involving high-molecular-weight block polymers (BCPs), star-shaped polymers, surface modification, and biomedicine due to properties like high efficiency, speed, and reasonably good modularity [16,17,18,19,20,21,22,23,24,25,26,27,28,29,30,31,32]. The extensively researched RAFT-HDA reaction is based on several relatively electron-rich dithioesters, including diethoxyphosphoryl dithiocarbonate (BDEPDF) and benzyl pyridin-2-yl dithiocarbonate (BPDF), which act as dienophiles and can effectively close the HOMO-LUMO gap in the DA reaction due to the electron-absorbing groups, leading to efficient and quick DA reactions. Cyclopentadienes and linear dienes make up the majority of all dienes. Manufacturing dienes often needs several reaction steps to obtain the building blocks with dienes as end groups for future reactions. Topological structures with higher complexity are characterized by more difficult steps and lower efficiency [33,34,35,36].

Microgels have high water content, biocompatibility, and acceptable chemical and mechanical properties with changeable dimensions. They are functionalized nanoparticles with an internal three-dimensional cross-linked network. Applications in drug delivery, bioimaging, and other fields (coatings, catalysis, and photonics) are made possible by the material’s responsiveness to external stimuli, such as temperature, pH, light, electric field, and ionic strength [37,38,39,40,41,42,43]. Microgels can be divided into two kinds based on the molecular network structure: covalent cross-linking and supramolecular cross-linking. The main covalent cross-linking techniques are free radical polymerization, click chemistry, Schiff base reaction, thiol–disulfide exchange, and light reaction. Covalent cross-linking generally necessitates the introduction of various cross-linking agents into the microgel system to prepare precursors with various reactive functional groups. On the one hand, introducing cross-linking agents can result in toxicity. Still, on the other hand, the preparation of microgels and the regulation of various properties necessitates more complex synthesis steps. Microgels formed by supramolecular cross-linking are based on the self-assembly polymerization of various physical interactions (including ionic, hydrophobic, and hydrogen bonds). They are typically prepared under mild conditions, primarily in water, to avoid toxic effects [44,45,46,47,48,49,50,51]. Gels formed by supramolecular cross-linking, on the other hand, may have lower mechanical strength and stability [52,53,54,55,56,57]. Covalently cross-linked microgels are formed by coupling reactive functional groups, allowing for the modulation of the structure and properties of microgel particles with colloidal stability, which is required to prevent the gel network from dissociating and leaking drugs. To prepare microgels with dynamic covalent bonding capable of maintaining strength while achieving functional flexibility, efforts to develop stable microgels with simple, efficient, and mild reaction conditions are critical.

Herein, we achieved ultrasound-responsive properties in microgels and the release of model drug molecules using a simple preparation of responsive microgels based on RAFT-HDA dynamic covalent bonding. Firstly, the polymerization of styrene mediated by the RAFT reagent with double-ended BDEPDF as a double-ended dienophile linker and the copolymerization of furfuryl methacrylate (FMA) with styrene monomer to obtain a chain were achieved [58], and a pendant side group was used to realize the successful occurrence of the DA reaction between phosphoryl disulfide and furan. Following that, the microgel system was created via HDA reaction between phosphoryl disulfide and furan groups, with the dienophile building block being a copolymer of poly (ethylene glycol monomethyl ether acrylate) (OEGMA) and FMA and the cross-linker and hydrophobic segment being a polystyrene dienophile linker. We investigated the effects of the single-molecule chain functional group degree (f), furan/dithioester ratio (r), and polystyrene linker length (D) on the DA reaction and microgel system by varying the number of furans in the dienophile chain and the length of the dienophile linker. On the other hand, since RAFT polymerization has good end-group retention and controllable molecular weight, the effects of the single-molecule chain functional group degree (f), furan/dithioester ratio (r), and polystyrene linker length (D) on the DA reaction and consequently on the microgel system were investigated by varying the number of furans in the dienophile chain and the length of the dienophile linker. These microgels can release model drug molecules under regulated conditions, and their mild, easy, and effective production processes are anticipated to have significant promise for use in areas like biomedical materials.

## 2. Results and Discussion

The main working concept is illustrated in Figure 1a. We prepared polystyrene with phosphoryl disulfide caps as the diene and a copolymer of FMA and styrene as the dienophile to test the viability of the HDA reaction between the phosphoryl disulfide and furan ring. The original concept of Figure 1a was later expanded to include the creation of microgels by forming copolymers of FMA and OEGMA as in Figure 1b.

In the microgel system, polystyrene, a dienophile-type double-ended cross-linker, was subjected to an HDA reaction with OEGMA-co-FMA to produce microcross-linked block polymers, in which polystyrene served as a hydrophobic segment and OEGMA as a hydrophilic segment. Following the HDA reaction, the system’s hydrophilic balance was further assembled in an aqueous solution to produce the microgel.

A more detailed description can be found in the Appendix A. Polystyrene (PS1–PS4) was created by RAFT polymerization, adopting 1,4-phenyl bis (methylene) bis((diethoxyphosphoryl) methyl dithiocarbonate) as a chain transfer agent as the anticipated dienophile molecule in the HDA model concept (Appendix A). Simultaneously, 4-cyano-4-(dodecyl trithiocarbonate)pentanoic acid (DDTCP) was used to prepare RAFT copolymers of furfuryl methacrylate and styrene (SF1–SF2) (Appendix A) or polyethylene glycol monomethyl ether acrylate (OF1-OF4) (Appendix A). 

The produced polymers were evaluated by ^1^H NMR analysis to ascertain the structure of the above polymers. All of the distinctive peaks of the RAFT reagent, dienophile, and pro-dienophile building blocks can be seen in Figure 1. As seen in Figure 1a, the characteristic peaks in proton signals b and d corresponding to phosphoryl disulfide show that the P-Di-linker successfully mediated the polymerization of styrene, resulting in the presence of a well-defined disulfide end group. In addition, the characteristic peaks in proton signals A, a, C, D, c, and d correspond to styrene, ethylene glycol, and furan. To investigate the feasibility of the reaction of phosphoryl dithioesters with furan HDA, two sets of dienophile building blocks of different lengths and different numbers of furan units, SF1, are characterized by several different means.

In the ^1^H NMR analysis (Figure 1c), the proton signal f corresponds to the formation of new double-bonded hydrogen characteristic peaks after the cycloaddition reaction of the furan ring with the dithioester ring. Figure 2a,b depict the relationship between the molecular weights of the two sets of reaction building blocks and the microcross-linked polymers. The SEC chromatograms show the traces of the formed microcross-linked polymers relative to the individual building blocks. The significantly reduced retention time of the traces and the absence of significant shoulders also indicate successful and effective crosslinking formation, which is consistent with the data in Table 1. It should be emphasized that none of the microcross-linked polymers produced in the current investigation were pure when they underwent SEC testing. We can infer from the SEC spectra of the microcross-linked polymers that the grafting rate of the system, or the effectiveness of the HDA reaction, is influenced by the length of the block and the number of furan units in a single diene building block. This, in turn, results in a variety of molecular weight distributions for the system. This issue has not been thoroughly explored yet, and the microgel part requires further analysis.

DSC and variable-temperature FTIR were applied to examine further the thermal reversibility of the Diels–Alder reaction in the system. It is possible to visualize the system after the HDA reaction in the DSC curve with a heat absorption peak at approximately 120 °C (Figure 3b), which corresponds to the retro-DA reaction of the DA bond between the furan and dithioester. Also, it supports the viability of the HDA reaction between the furan ring and phosphoryl dithioester from the side. The microcross-linked polymers were analyzed by variable temperature FTIR spectroscopy in addition to DSC (Figure 3a). The technique is highly sensitive to the exact molecular structures created by the Diels–Alder and retro-Diels–Alder processes being absorbed. The samples were heated to a temperature over the 120 °C threshold for the retro-DA reaction, a gradient of 20 °C was set, and each temperature was kept for 10 min before the corresponding analysis, which consisted of heating and cooling for one cycle, was conducted. With symmetric and asymmetric stretching vibrations of the intra-ring C-O-C bonds, furan rings have the cyclic bis(vinyl ether) structure typical of these compounds. Asymmetric stretching vibrations of the ring’s C-O-C bonds occur in furan acrylates at a frequency of 1220 cm^−1^. The cyclic C-O-C bond’s asymmetric stretching vibration occurs at a frequency of 1220 cm^−1^ in furan acrylate, while its symmetric stretching vibration occurs at a reasonably steady frequency of 1076 cm^−1^ and is less affected by the substituent (IR analysis of the structure and IR spectral features of α-furan esters). The characteristic peaks at 1030 cm^−1^ and in the range of 1750–2400 cm^−1^ vary in the spectra of the microcross-linked polymers. The analysis shows the stretching vibration of the C-O-C bond, which is responsible for the absorption band at 1030 cm^−1^, the stretching vibration of the carbonyl group, which is responsible for the absorption band at 1750 cm^−1^, and the variation in the absorption intensity of the C=C=O bond, which is responsible for the absorption band at 2400–2150 cm^−1^.

As can be seen, as the temperature rises, the system experiences a retro-DA reaction, and the absorption intensities of the relevant IR characteristic peaks vary accordingly. However, the system’s recyclability could be better. Nevertheless, the feasibility of the HDA reaction in this system can be confirmed, and Appendix A further shows that the system is reversible. The typical TGA curves of the pro-dienophile building block PS, dienophile building block SF, and SF-S crosslinking reaction are shown in Figure 3c. As observed, including FMA lowers the temperature at which breakdown begins but raises the quantity of carbon still present at 700 °C. A greater amount of residual char was left after the cross-linking with PS, proving that the cross-linking was successful. The degree of cross-linking and the amount of residual char are directly connected, and the relative amount of FMA is proportional to both (Appendix A). It is important to note that the stability of the system decreased after the reaction, which can be attributed to the further increase in the amount of FMA after the grafting of the system, that the cross-linked system did not form a large cross-link density, and that the reaction system was not purified and there were unreacted building blocks.

As a result of the research above, we expanded the DA reaction between furan and phosphoryl disulfide to create a microgel system. The number of functional groups (f) of the single-chain polystyrene was fixed at 2, and the disulfide ester and furan ring existed here as cross-linking reaction sites. The number of furan sites of the single-chain furan copolymer can be adjusted, but it is not advisable to have too many, as cross-linking reaction sites form for creating a microgel system. At the same time, OEGMA and polystyrene interact to generate hydrophilic linkage segments and comb trapezoidal or smaller cross-link density block polymers, depending on the molecular structure of the copolymer. Here, we used adjustable RAFT polymerization to create microgels more easily while examining the impact of the hydrophobic segment length and cross-link density on the microgel system.

We first prepared polystyrene of three different lengths (DP 30, 40, and 80, respectively) of S2, S3, and S4 as cross-linkers to investigate the impact of hydrophobic chain segment length on microgels. We then obtained a series of solutions containing the S/OF reaction system by controlling the dithioester to furan ratio of 1 (r = 1), reacting with OF4 via HDA, and performing subsequent microgel preparation (for details, see the Section 3). The size and shape of the grafted polymer aggregates were investigated using transmission electron microscopy (TEM). The TEM morphology supported our prior theory that the polymers generated from the S/OF process could form microgel structures in water, as illustrated in Appendix A. Moreover, the TEM images and DLS data (Figure 4) show that the size of the microgel gradually increased with the length of the hydrophobic segment. This is explained by the fact that as the hydrophobic part of the system grew, the system had to reach a new affinity for water, causing a corresponding change in the microgel size. The cross-linking density of the microgel system, which had varied impacts on the particle size and dispersion of the microgel, was also affected by the size of the building block for the DA reaction.

The control group (Figure 5) with varying water concentrations (40–80%) demonstrates that the water content did not affect the microgels’ fundamental spherical form. We refer to this as forming a microcross-linked network structure that was difficult to change, coupled by DA bonds between the hydrophobic chain segment S and the hydrophilic chain segment OF. According to the TEM images, however, numerous stiff benzene rings on the molecular weight also contributed to the production of a few components, such as a bowl-shaped morphology and porous morphology, because the hydrophobic chain segment of the system was polystyrene. This issue requires additional research and is not further covered here. The DLS data show that the microgel’s particle size fluctuated with increasing water content but did not change appreciably either way. 

In subsequent experiments, we effectively controlled the number of cross-linking sites by regulating the number of furan groups in single chains via RAFT polymerization regulation, aiming to control the cross-linking density of the cross-linked system. To investigate the effect of cross-linking density on the microgel system, we prepared OEGMA-co-FMA block copolymers OF1, OF2, and OF3 (=2, 4, 7) containing varying numbers of furan units while keeping the length of hydrophobic and hydrophilic chains constant. The morphology of the resulting microgels composed of S3 and OF with different cross-linking densities is shown in TEM images after a similar preparation process (Figure 6). The microgels of various cross-linking densities had a spherical shape. As the number of single-chain furan units rose, the system’s cross-linking density and the number of linked hydrophobic chain segments also grew. This resulted in a general increase in gel size and a corresponding narrowing of the particle size distribution.

Notably, the research on both the hydrophobic chain length and the number of single-chain furan units relies on the same initial functional group concentration and a dithioester/furan ratio of 1 ([S]/[F] = r = 1). Next, we investigated how parameter r affected the microgel system. Using r = 2 for the corresponding reaction process, we found that too much furan caused the system to change to gel, resulting in an insoluble fraction. Only a small amount of soluble fraction was needed for the corresponding characterization, proving that the large aggregated system was not a properly prepared microgel (Appendix A). The microgels made via this method also exhibited good stability; just a small amount of precipitation was discovered in the system after 36 days in a normal environment, and the solution of the microgel system changed from pure white to translucent white. The accumulation was more significant.

Microgels have the potential for biomedical applications such as controlled release because of their response to external stimuli. At the moment, temperature and pH variations are the main focus of the microgels’ responsiveness. The microgel system’s responsiveness to temperature is often constrained by the type of polymer used and depends on its LCST properties. Manufacturing microgels with physically cross-linked structures necessitates using unique molecules and structures. This is a dilemma because the reaction to pH typically depends on specific bonding, such as hydrogen bonding.

According to the above findings, the DA bond created by the RAFT-HDA reaction was thermally reversible up to 120 °C. Even though the DA bond would break to stimulate the microgel responsiveness, the microgel at this temperature undoubtedly disintegrated due to the high temperature, failing to match the application requirements of the microgel. In 2014, Wang et al. [59] created a brand-new mechanical carrier made at room temperature; this polymer could undergo a retro-HDA reaction through ultrasonication, causing the system’s disintegration and giving it force-responsive characteristics. Inspired by the above findings, we researched the force responsiveness of the HDA bond between the furan ring and phosphoryl dithioester. We chose two systems, OF3-S2 and OF4-S2, for 2 h of sonication to test the microgel systems’ force responsiveness. Figure 7a,b show the phenomena of polystyrene molecules separating from the microgels and generating flocculent precipitates. Gel chromatography analysis revealed that the molecular weight of the sonicated systems significantly decreased, and lower peaks appeared in the higher-molecular-weight fraction corresponding to the non-detached fraction (Figure 7c,d). The analysis was subsequently expanded upon by coating rhodamine B.

Rhodamine B was coated by the microgel technique at a comparatively high rate, as seen in Figure 8. The bonding of the microgel system was broken after sonication, reducing the pace at which the medication was coated. The change in the size of the microgel particles was also noticeably smaller after sonication. The mechanical force that disrupts the bonding between the polystyrene and copolymer chain segments, which results in the microgel system’s disruption and gives the microgel force responsiveness, is thought to be responsible for the results mentioned above. However, further research must determine the precise mechanism and contributing elements.

## 3. Materials and Methods

1,4-bis(bromomethyl)benzene (97%, Aladdin), sodium hydride (60% dispersion in mineral oil, Aladdin), tetrahydrofuran (THF) (anhydrous, ≥99.9%, J&K), diethyl phosphate (99.0%, Aladdin), carbon disulfide (anhydrous, ≥99.9%, Aladdin), toluene(Macklin), chloroform(J&K), dimethyl sulfoxide (DMSO)(J&K), styrene(Amethyst, ≥99.5%), silica gel(Aladdin), polyethylene glycol monomethyl ether acrylate(OEGMA)(Mn = 480, Aladdin, ≥99%), furfuryl methacrylate(≥95%, J&K), 2,2″-azobisisobutyronitrile (AIBN)(J&K), and alkaline alumina(200–300 order, Macklin) were used. The RAFT agents 1,4-phenylene bis (methylene)bis((diethoxyphosphoryl)methanedithioformate) (P-Di-linker) and 4-cyano-4-(dodecyl trithiocarbonate)pentanoic acid(DDTCP) were prepared according to published procedures (see Appendix A).

### 3.1. Characterizations

^1^H nuclear magnetic resonance (NMR) spectroscopy was performed on an AVANCE III 400 spectrometer operating at 400 MHz for hydrogen nuclei. All samples were dissolved in either C*D*Cl_3_ or DMSO-*d*6.

A size exclusion chromatography (SEC) system equipped with 10 μm mixed columns in series and a line with a 20 A refractive index detector was used. The mobile phase of the instrument was *N*, *N*-dimethylacetamide (DMAC), the flow rate was 1 mL∙min^−1^, the temperature of the gel chromatographic column rose to 50 °C, and the standard curve was made by testing polystyrene with different molecular weights.

Light scattering (DLS) performed on a Malvern Zetasizer Nano series (Nano-ZS) instrument was used to measure the microgel sizes. The samples were illuminated with a 633 nm He–Ne laser, and the scattering light at a 90° angle was recorded using an avalanche photodiode detector.

An Hitachi HT7700 transmission scanning electron microscope (TEM) was used to observe the morphology of the microgels with an observation voltage of 120 kV.

### 3.2. Synthesis of Phosphoryl Disulfide Terminated Polystyrene (PS1-PS4)

A mixture of styrene monomer (in advance with alkaline alumina to remove the polymerization inhibitor), azo diisobutyronitrile (AIBN), and P-Di-linker was prepared in a Schlenk tube.

Oxygen and water were removed from the system through three freeze-evacuate-thaw cycles. The reaction was carried out at 75 °C for 12 h and then stopped by placing the solution in an ice water bath and exposing it to oxygen. Cold methanol precipitated the product mixture thrice, and the mixture was dried in a vacuum oven to obtain the corresponding products. The details of the four polymers were as follows (PS1-[M]_0_/[RAFT]_0_/[AIBN]_0_ = 300:1:0.2, Mn = 6689 g∙mol^−1^, *Ð* = 1.15, conversion = 21%; PS2-[M]_0_/[RAFT]_0_/[AIBN]_0_ = 145:1:0.2, Mn = 2832 g∙mol^−1^, *Ð* = 1.15, conversion = 18.86%; PS3-[M]_0_/[RAFT]_0_/[AIBN]_0_ = 200:1:0.2, Mn = 3987 g∙mol^−1^, *Ð* = 1.4, conversion = 19.2%; and PS4-[M]_0_/[RAFT]_0_/[AIBN]_0_ = 400:1:0.2, Mn = 7563 g∙mol^−1^, *Ð* = 1.13, conversion=18.2%).

### 3.3. Synthesis of a Long Chain with Furan as the Group Vertical Side Group

A mixture of styrene monomer or OEGMA, furfuryl methacrylate monomer (in advance with alkaline alumina to remove the inhibitor), azo diisobutyronitrile (AIBN), and DDTCP was prepared in a Schlenk tube. Oxygen and water were removed from the system by three freeze-evacuate-thaw cycles. The reaction was carried out at 75 °C for 12 h or 5 h and then stopped by placing the solution in an ice water bath and exposing it to oxygen. The copolymers of styrene and FMA were precipitated three times with cold methanol, and the copolymers of OEGMA and FMA were precipitated three times with cold hexane, followed by drying in a vacuum oven to obtain the corresponding products. The details of several polymers were as follows:(SF1-[S]_0_/[F]_0_/[RAFT]_0_/[AIBN]_0_ = 150:44:1:0.2, conversion^a^(conversion rate of the previous component) of 40%, conversion^b^(conversion rate of the previous component) = 79.5%,[S]/[F] = 60:35; SF2-[S]_0_/[F]_0_/[RAFT]_0_/[AIBN]_0_ = 115:20:1:0.2, conversion^a^ = 37.4%, conversion^b^ = 84.6%, [S]/[F] = 43:17; OF1-[O]_0_/[F]_0_/[RAFT]_0_/[AIBN]_0_ = 55:5:1:0.2, conversion^a^ = 81.8%,conversion^b^ = 40%, [O]/[F] = 45:2OF2-[O]_0_/[F]_0_/[RAFT]_0_/[AIBN]_0_ = 55:10:1:0.2, conversion^a^ = 81.8%, conversion^b^ = 40%,[O]/[F] = 45:4; OF3-[O]_0_/[F]_0_/[RAFT]_0_/[AIBN]_0_ = 55:15:1:0.2, conversion^a^ = 81.8%, conversion^b^ = 46.7%, [O]/[F] = 45:7; OF4-[O]_0_/[F]_0_/[RAFT]_0_/[AIBN]_0_ = 55:10:1:0.2, conversion^a^ = 72.7%, conversion^b^ = 50%, [O]/[F] = 40:5).

### 3.4. Reversible Addition Fragmentation Chain Transfer Hydro Diels–Alder Cyclization between Phosphoryl Disulfides and Furan Rings

The phosphoryl dithioester capped polymer and the pendant furan ring long chain polymer were dissolved in toluene in a glass bottle with stirring and heat at 55 °C for 24 h. The resulting crosslinked polymers were separated by precipitation in cold hexane and analyzed by SEC (THF or DMAC) and NMR (C*D*Cl_3_).

### 3.5. Preparation of Polymeric Microgels

By pipetting, toluene was added to 20 μL of the reaction mixture and diluted to 0.6 mL; the diluted mixture was stirred for 10 min. Ultra-pure water was then added dropwise to the mixture using a peristaltic pump at a rate of 0.67 mL∙h^−1^ until the desired water–toluene volume ratio was obtained. The mixed solution was then placed into a cellulose dialysis strip, and the toluene and unreacted building blocks were removed by ultra-pure water dialysis (water changed every 3 h, at least five times). The solutions were subsequently used for TEM and DLS characterization. 

## 4. Conclusions

Based on the cascade reaction of RAFT polymerization and Diels–Alder, this paper investigated a new diene furan ring for the HDA reaction and applied the new RAFT-HDA reaction to the preparation of microgels. On the one hand, the furan ring could participate in the HDA reaction, providing a more readily available dienophore for the RAFT-HDA reaction, expanding the variety of dienophores in the RAFT-HDA reaction system, and making it easier to obtain more complex topologies from RAFT-HAD. On the other hand, the RAFT-HDA reaction could be extended to some extent for more applications. In this paper, we investigated the effects of hydrophobic chain length and cross-linking density on microgel systems, obtaining stable microgel assemblies in a very simple way while achieving easy modulation of their properties due to the introduction of controllable RAFT polymerization, which made them mechanically responsive. The presence of furan rings in many monomers and their investigation as novel dienes gave more possibilities for the RAFT-HDA reaction. Furan rings were found in many monomers, and researching them as novel dienes expanded the RAFT-HDA reaction’s potential.

## Data Availability

The data presented in this study are available in article and Appendix A.

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
