# Peer review of "Responsive Microgels through RAFT-HDA Dynamic Covalent Bonding Chemistry"

_molecules, 2024, doi:10.3390/molecules29061217_

Round 1
Reviewer 1 Report
Comments and Suggestions for Authors
The Manuscript from Yin and coworkers reported the synthesis and characterization of smart microgels through RAFT-HAD. Generally speaking, this work is interesting, especially for these who are interested in developing intelligent microgels via convenient polymerization protocols. However, this manuscript requires substantial revision prior to be accepted for publishing. Specifically,
(1) The title is somehow misleading, and I would suggest changing to “Responsive microgels through RAFT-HDA chemistry”, to avoid the repetition. The corresponding part in the abstract and main text should also be changed.
(2) According to IUPC, gel permeation chromatography (GPC) should be substituted by size exclusion chromatography (SEC).
(3) Microgel fabrication: what’s the effects of copolymer concentrations? How about the controllability of the sizes of the microgels?
(4) How to control the crosslinking degree for the microgel fabrication?
(5) Microgel formation through assembly: what ‘s this really means? i.e. authors should provide experimental data to support their point view that the microgels formed via assembly. I would prefer to say the microgels were formed just through crosslinking, accompanied with aggregation. If one wants to say the aggregation is assembly, detailed experimental results are required.
(6) Composition of the copolymers can be obtained from 1H NMR spectra recorded at elevated temperatures to increase the signal resolution.
(7) Legends in Figure 1 and 2 should be revised. In Figure 1, the legends should contain necessary information for each image. For legend of Figure 2, one should describe these as SEC elution curves, instead of the discussions written in the legend.
(8) Legend in Figure 3 is confusing. Please revise it.
(9) Legend in Figure 4 is not proper, since DLS never can provide image, but only curves. (a), (b) and (c) should be clearly indicated what they are. Similar problem existed in Figure 6, 7 and 8.
(10) Legends in Figure 4 and 5 should be changed into “…microgels from copolymers with different lengths of hydrophobic segment”.
Comments on the Quality of English LanguageEnglish writing of the whole manuscript should be significantly revised. For example, in page 2 and 3, line 90-98:
"We investigated the effects of single-molecule chain functional group degree (f), furan/dithioester ratio (r), and polystyrene linker length (D) on the DA reaction and thus on the microgel system by varying the number of furans in the dienophile chain and the length of the dienophile linker on the DA reaction and thus on the microgel system. We investigated the effects of single-molecule chain functional group degree (f), furan/dithioester ratio (r), and polystyrene linker length (D) on the DA reaction and consequently on the microgel system by varying the quantity of furans in the dienophile chain and the length of the dienophile linker because RAFT polymerization has good end-group retention and controllable molecular weight."
Author Response
Dear Reviewer,
First and foremost, I would like to express my deepest gratitude to you for taking the time out of your busy schedule to review my manuscript and provide a series of insightful and constructive comments. I fully understand that these suggestions are crucial in enhancing the quality of my research work and the scholarly value of my manuscript.
After carefully considering each of your review points, I have made corresponding revisions and improvements to my manuscript. In the following sections, I will address each of your comments individually, providing detailed explanations of the changes made or offering relevant discussions and justifications:
Comments (1) The title is somehow misleading, and I would suggest changing to “Responsive microgels through RAFT-HDA chemistry”, to avoid the repetition. The corresponding part in the abstract and main text should also be changed.
Re:Thanks for your kind suggestion. The reaction we use is the RAFT-HDA reaction, which forms a dynamic covalent bonding. And dynamic covalent bonding is the reason for producing responsive special properties. Therefore, we believe that the keywords RAFT-HDA and dynamic covalent bonding cannot be ignored. Based on your suggestion, we have changed the title to “Responsive microgels through RAFT-HDA dynamic covalent bonding chemistry”.
Comments (2) According to IUPC, gel permeation chromatography (GPC) should be substituted by size exclusion chromatography (SEC).
Re:Thanks for your kind suggestion. According to your suggestion, the gel permeation chromatography (GPC) in the article has been substituted by size exclusion chromatography (SEC)
Comments (3) Microgel fabrication: what’s the effects of copolymer concentrations? How about the controllability of the sizes of the microgels?
Re:Thanks for your kind suggestion. The concentrations of polymer generating microgel are shown in Table 1, among which OF-S series are components forming microgel. We can form microgel in the concentration range of 0.06-0.12. At the same time, because of the traditional assembly method, the polymer concentration cannot be too high, otherwise, aggregation and precipitation will occur, and the balance of hydrophilic and hydrophobic segments cannot occur.
Meanwhile, the size of microgel is controllable. We have verified this conclusion in repeated experiments, and it is worth mentioning that its size can change with the change of water content (Figure 5), which provides more possibilities for its application.
Comments (4) How to control the crosslinking degree for the microgel fabrication?
Re:Thanks for your kind suggestion and sorry for possible misunderstandings. The degree of crosslinking of microgel is controlled by RAFT-HDA reaction. Specifically, it is achieved by controlling the density of furan rings in the polymer chain (blue in Figure 1). Polymers (OF1-4) with different furan ring densities can form microgels with different crosslinking degrees.
Comments (5) Microgel formation through assembly: what ‘s this really means? i.e. authors should provide experimental data to support their point view that the microgels formed via assembly. I would prefer to say the microgels were formed just through crosslinking, accompanied with aggregation. If one wants to say the aggregation is assembly, detailed experimental results are required.
Re:Thanks for your kind suggestion. Self-assembly of amphiphilic polymers to form micro gel is a widely reported method, such as references 41, 42, etc. Among them, the formation of cross-linking point is an important prerequisite for gel. In the "aggregation" process you mentioned, due to the difference in solubility of hydrophilic and hydrophobic polymer segments in cross-linked polymers, automatic aggregation occurs, often forming size stable nanostructures, which is what we call self-assembly process.
Comments (6) Composition of the copolymers can be obtained from 1H NMR spectra recorded at elevated temperatures to increase the signal resolution.
Re: Thank you for your very valuable suggestion. We tried the heating nuclear magnetic resonance method, but unfortunately, we did not obtain data better than Figure 1. In Figure 1, the degree of separation of characteristic peaks is good, which allows us to integrate calculations to obtain the composition of the polymer.
Comments (7) Legends in Figure 1 and 2 should be revised. In Figure 1, the legends should contain necessary information for each image. For legend of Figure 2, one should describe these as SEC elution curves, instead of the discussions written in the legend.
Re: Thanks for your kind suggestion. According to your suggestion, the legend in Figure 1 has been revised to necessary information for each image, the legend in Figure 2 has been revised to SEC evolution curves.
Comments (8) Legend in Figure 3 is confusing. Please revise it.
Re: Thank you for your suggestion. We have revised the captions of the article based on your feedback.
Comments (9) Legend in Figure 4 is not proper, since DLS never can provide image, but only curves. (a), (b) and (c) should be clearly indicated what they are. Similar problem existed in Figure 6, 7 and 8.
Re: Thanks for your kind suggestion. According to your suggestion, the legends and corresponding content in the article have been modified.
Comments (10) Legends in Figure 4 and 5 should be changed into “…microgels from copolymers with different lengths of hydrophobic segment”.
Re: Thanks for your kind suggestion. According to your suggestion, the legends in Figure 4 and 5 have be changed into “…microgels from copolymers with different lengths of hydrophobic segment”.
Comments (11) Comments on the Quality of English Language
English writing of the whole manuscript should be significantly revised. For example, in page 2 and 3, line 90-98:
"We investigated the effects of single-molecule chain functional group degree (f), furan/dithioester ratio (r), and polystyrene linker length (D) on the DA reaction and thus on the microgel system by varying the number of furans in the dienophile chain and the length of the dienophile linker on the DA reaction and thus on the microgel system. We investigated the effects of single-molecule chain functional group degree (f), furan/dithioester ratio (r), and polystyrene linker length (D) on the DA reaction and consequently on the microgel system by varying the quantity of furans in the dienophile chain and the length of the dienophile linker because RAFT polymerization has good end-group retention and controllable molecular weight."
Re: I appreciate your comments on the quality of English language in my manuscript. I understand that clear and accurate language is crucial for ensuring that the research is understood. And I recognize that there are areas where I may have fallen short in expressing myself clearly or precisely in English as a non-native speaker. To address these concerns, I have carefully reviewed and revised the manuscript, focusing on improving the language quality. I am confident that these revisions will significantly improve the readability and comprehensibility of my manuscript.
Thank you once again for your attention and guidance on my manuscript. I look forward to receiving further feedback from you.
Best regards,
Author

Reviewer 2 Report
Comments and Suggestions for Authors
The article discusses the synthesis of microgels via RAFT polymerization. Three different polymer compounds were synthesized where microgel formation was achieved over self-assembly of those compounds. Such microgels can absorb and release Rhodamine molecules under sonication. The overall manuscript is well written. The presented Scheme clearly depicts the strategy. I cannot comment too much about the Scientific quality of the synthesis and analysis, due to not my major field However, there is still plenty of small points and comments, some more technical, which could be improved the quality of the article. These are mentioned in bullet points as follows:
1. Authors use as a multiplication sign “.” Instead of “∙”, which I saw in almost every image, text and tables (for example table 1). Please use the right sign to it in the entire manusscript.
2. Figure 1: I recommend labelling the compounds in the image with abbreviation number typically discussed in the article. For a non-synthetic person this is otherwise hard to track.
3. Line 150-156 Check text, there are many missing spaces. For example, “conversionb”
4. Figure 2: Increase the DPI value and the legend labeling. The latter is barely readable.
5. Figure 3: Data in a is difficult to read out. Why not extend/stretch that image over (b) and (c) horizontally. The insets are too small and must be increased. The legend has double labelling with different color codes, thus unclear what does that mean. In (b) In the Y-Axis check for sign “.” and change to “∙” Increase the overall DPI value of the image
6. Figure 4: (d) Figure caption à Authors have written DLS images…. Better rephrase like that: Hydrodynamic diameter from DLS measurement as a function of… . In the image: Size (d.nm), maybe change to size d (nm) ?
7. Figure 5 have very unprecise figure caption. Add (a), (b), etc… into the image and label correctly every image. Also see for DLS data the Point 6.
8. Figure 8. (a,b) Are these UV-vis spectra or Emission spectra? Please label it correctly. (c,d) see for this figure caption also point 6. I guess these are DLS data.
9. It seems to be, that sonication completely destroys the microgel network. Not unexpected, due to the structure results from self-assembly and high energy yield to rapture. DLS data confirm the rapture. Maybe add microscopic images before and after sonication.
10. The article can improve the quality by adding measurements, to show the rapture tendency as a function of sonication time by displaying DLS data and microscopic images. Here one can iterate small changes in time within the range 0-2h.
11. Check the article on typos, there plenty of them.
Author Response
Dear Reviewer,
First and foremost, I would like to express my deepest gratitude to you for taking the time out of your busy schedule to review my manuscript and provide a series of insightful and constructive comments. I fully understand that these suggestions are crucial in enhancing the quality of my research work and the scholarly value of my manuscript.
After carefully considering each of your review points, I have made corresponding revisions and improvements to my manuscript. In the following sections, I will address each of your comments individually, providing detailed explanations of the changes made or offering relevant discussions and justifications:
Comments (1) Authors use as a multiplication sign “.” Instead of “∙”, which I saw in almost every image, text and tables (for example table 1). Please use the right sign to it in the entire manuscript.
Re: Thank you for your kind suggestion. The error has now been corrected, and all instances of "." have been replaced with "∙" as the multiplication sign throughout the entire manuscript, including in the images, text, and tables.
Comments (2) Figure 1: I recommend labelling the compounds in the image with abbreviation number typically discussed in the article. For a non-synthetic person this is otherwise hard to track.
Re: Thank you for your kind suggestion. We have revised Figure 1 based on your feedback.
Comments (3) Line 150-156 Check text, there are many missing spaces. For example, “conversionb”
Re: Thank you for your kind suggestion. Thank you for pointing out this issue in lines 150-156. The inaccurate content in the text has been corrected, such as “conversionb”, which has been changed to “conversionb”.
Comments (4) Figure 2: Increase the DPI value and the legend labeling. The latter is barely readable.
Re: Thank you for your kind suggestion. As your suggestion, the legends in Figure 2 are not clear enough to accurately express the content. DPI value and the legend labeling have been added as required in Figure 2.
Comments (5) Figure 3: Data in a is difficult to read out. Why not extend/stretch that image over (b) and (c) horizontally. The insets are too small and must be increased. The legend has double labelling with different color codes, thus unclear what does that mean. In (b) In the Y-Axis check for sign “.” and change to “∙” Increase the overall DPI value of the image.
Re: Thank you for your kind suggestion and sorry for the trouble it has brought you . We have revised Figure 3 based on your feedback, especially the captions on Figure 3a make it easier for people to understand and see.
Comments (6) Figure 4: (d) Figure caption à Authors have written DLS images…. Better rephrase like that: Hydrodynamic diameter from DLS measurement as a function of… . In the image: Size (d.nm), maybe change to size d (nm) ?
Re: Thank you for your kind suggestion. We have revised the Figure 4 and corresponding text description.
Comments (7) Figure 5 have very unprecise figure caption. Add (a), (b), etc… into the image and label correctly every image. Also see for DLS data the Point 6.
Re: Thank you for your kind suggestion. Figure 5 has been rearranged and designed to make it easier to understand.
Comments (8) Figure 8. (a,b) Are these UV-vis spectra or Emission spectra? Please label it correctly. (c,d) see for this figure caption also point 6. I guess these are DLS data.
Re: Thank you for your kind suggestion. We have revised the Figure 8 and corresponding text description.
Comments (9) It seems to be, that sonication completely destroys the microgel network. Not unexpected, due to the structure results from self-assembly and high energy yield to rapture. DLS data confirm the rapture. Maybe add microscopic images before and after sonication.
Re: Thank you for your kind suggestion. As you said, ultrasound will completely destroy the micro gel structure. As shown in Figure 1, the bond of ultrasonic response is on the cross-linking site, which leads to the complete collapse of the micro gel structure. We did conduct TEM tests before and after ultrasound, and the results were similar to those in Figures 4-6; After ultrasound, only agglomerated polymers can be observed without a regular structure, so it is not considered to have reference value and is not supplemented here.
Comments (10) The article can improve the quality by adding measurements, to show the rapture tendency as a function of sonication time by displaying DLS data and microscopic images. Here one can iterate small changes in time within the range 0-2h.
Re: Thank you for your kind suggestion. In the main text, we did only present the state of 2h; We conducted additional tests for this and found that such changes do not vary linearly with ultrasound time. Among them, we took samples and observed that the size of the micro gel in the system almost did not change significantly (as shown in the figure). This also provides more possibilities for its controllable release as a guest molecule.
Comments (11) Check the article on typos, there plenty of them.
Re: I appreciate your comments on the quality of English language in my manuscript. I understand that clear and accurate language is crucial for ensuring that the research is understood. And I recognize that there are areas where I may have fallen short in expressing myself clearly or precisely in English as a non-native speaker. To address these concerns, I have carefully reviewed and revised the manuscript, focusing on improving the language quality. I am confident that these revisions will significantly improve the readability and comprehensibility of my manuscript.
Thank you once again for your attention and guidance on my manuscript. I look forward to receiving further feedback from you.
Best regards,
Author

Reviewer 3 Report
Comments and Suggestions for Authors
The authors decsribe a way of making microgels by taking advantage of a heter Diels Alder reaction, because of its reversibility. Polymers are made with thioester end groups serving as the dienophile and other polymers are made with pendant furan units serving as the diene. These aremixed and microgels are prepared in water containing media. The authors also do a preliminary investigation on the release of a dye by sonicating te gels and hereby brteaking the bonds. This part is rather superficially done and the idea of coating the particle with rgodamine is not well explained.
I have the follosing comments that need to be addressed before the paper can be accepted
1. inverse DA should be retro-Diles Alder as this is as far as I know the proper name
2. Introduction: I do not understand why iEDDA is mentioned, the authors do not use this reaction so I think it can be removed. As written now it works confusing
3.Explain all abbreviations especially in the abstract
4. The reaction between furan and maleimide is a traditional DA reaction and not a inverse electron demand Diels-Alder. The way this is written in the introduction is confusing
5. A Diels Alder reaction requires a diene and a dienophile. The statement “phosphoryl disulfide caps as the dienophile and a copolymer of FMA and styrene as the dienophile” can therefore not be correct
6. All systems made are in fact crosslinked networks, albeit of low crosslink density (2 endgroups of thioester for and several furan pendants for SF). Why can GPC still be measured?? And why do the authors refer to this as block copolymers?
7. Table 1: all abbreviations need to be defined (f, for example), and superscripts need explanation (a-b for example). So please add a legend to the Table
8. The procedure to prepare the microgels is rather brief (which reaction mixture? what is the concentration? how were the resulting solutions used for TEM and DLS, directly or were they further diluted??
9. Sonication can be brutal. How do the polymers before DA behave when sonicated, is MW also going down there?? Or does it remain stable. I think this is an important control reaction to add.
10. Figure 8 does not show the rate of coating, but are just DLS and UV traces. Please rephrase "comparativley high rate" as rate is not something that is measured in Fig 8.
Comments on the Quality of English LanguageEnglish can be improved, especially on page 11 bottom part (cursory research = curiosity research)
Author Response
Dear Reviewer,
First and foremost, I would like to express my deepest gratitude to you for taking the time out of your busy schedule to review my manuscript and provide a series of insightful and constructive comments. I fully understand that these suggestions are crucial in enhancing the quality of my research work and the scholarly value of my manuscript.
After carefully considering each of your review points, I have made corresponding revisions and improvements to my manuscript. In the following sections, I will address each of your comments individually, providing detailed explanations of the changes made or offering relevant discussions and justifications:
Comments (1) inverse DA should be retro-Diles Alder as this is as far as I know the proper name
Re: Thank you for pointing out the nomenclature issue regarding the retro-Diels-Alder. We apologize for any confusion caused by the incorrect terminology used in the previous context and have made the necessary corrections. Your attention to detail and expertise in this area are greatly appreciated.
Comments (2) Introduction: I do not understand why iEDDA is mentioned, the authors do not use this reaction so I think it can be removed. As written now it works confusing
Re: Thank you for raising your concern about the mention of iEDDA in the Introduction. Upon further review, we agree that the reference to iEDDA may be confusing. To avoid any potential confusion or misdirection, we have decided to remove the mention of iEDDA from the Introduction.
Comments (3) Explain all abbreviations especially in the abstract
Thank you for your feedback regarding the abbreviations used in our manuscript. According to the opinion, all abbreviations were supplemented with their full names when they first appeared.
Comments (4) The reaction between furan and maleimide is a traditional DA reaction and not a inverse electron demand Diels-Alder. The way this is written in the introduction is confusing
Re: We apologize for the misunderstanding caused. And inappropriate descriptions in the introduction have been modified.
Comments (5) A Diels Alder reaction requires a diene and a dienophile. The statement “phosphoryl disulfide caps as the dienophile and a copolymer of FMA and styrene as the dienophile” can therefore not be correct
Re: Thank you for pointing out this error. In a Diels-Alder reaction, there must be a distinction between the diene and the dienophile. In the statement, there is clearly a mistake since both compounds are being described as dienophiles. The statement “phosphoryl disulfide caps as the dienophile and a copolymer of FMA and styrene as the dienophile” has been modified to ”phosphoryl disulfide caps as the diene and a copolymer of FMA and styrene as the dienophile.” Similar content in the text has been rechecked.
Comments (6) All systems made are in fact crosslinked networks, albeit of low crosslink density (2 endgroups of thioester for and several furan pendants for SF). Why can GPC still be measured?? And why do the authors refer to this as block copolymers?
Re: Thank you for raising this question. As you mentioned, GPC characterization is an important part of this work. For highly cross-linked polymers, the cross-linked structure may cause the molecular chain to not be completely dissociated, thus affecting the measurement results of gel permeation chromatography. In cases where the crosslinks are small and the overall crosslink density is low, the polymer chains may still be able to pass through the pores of the chromatography column to some extent, allowing for GPC measurements.
In this work, the polymers have small molecular weights, and the density of the cross-linked structure was low. Therefore, we used GPC to analyze the polymer and obtained the results. We think the GPC results can prove the relationship between the molecular weights of the block copolymers and crosslinked networks.
As for the designation of the polymer, the naming of all block copolymers in this work has been changed to microcross-linked polymers as suggested.
Comments (7) Table 1: all abbreviations need to be defined (f, for example), and superscripts need explanation (a-b for example). So please add a legend to the Table
Re: Thank you for pointing out the need for clarity in Table 1. We agree that all abbreviations and superscripts should be defined to enhance readability and understanding. We have added a legend to the bottom of Table 1 explaining all abbreviations and superscripts used.
Comments (8) The procedure to prepare the microgels is rather brief (which reaction mixture? what is the concentration? how were the resulting solutions used for TEM and DLS, directly or were they further diluted??
Re:Thanks for your kind suggestion. The concentrations of polymer generating microgel are shown in Table 1, among which OF-S series are components forming microgel. We can form microgel in the concentration range of 0.06-0.12. At the same time, because of the traditional assembly method, the polymer concentration cannot be too high, otherwise, aggregation and precipitation will occur, and the balance of hydrophilic and hydrophobic segments cannot occur.
Meanwhile, the size of microgel is controllable. We have verified this conclusion in repeated experiments, and it is worth mentioning that its size can change with the change of water content (Figure 5), which provides more possibilities for its application.
Comments (9) Sonication can be brutal. How do the polymers before DA behave when sonicated, is MW also going down there?? Or does it remain stable. I think this is an important control reaction to add.
Re:Thanks for your kind suggestion. We strongly agree that this is a valuable controlled trial. First of all, the ultrasound we selected is relatively mild, which can be seen from the fact that it takes us 2 hours to destroy the micro gel. Based on our subsequent experiments, we found that ultrasound has no destructive effect on these C-C covalently bonded polymers, and their molecular weight can still be maintained.
On the other hand, we conducted additional tests for and found that such size changes do not vary linearly with ultrasound time. Among them, we took samples and observed that the size of the micro gel in the system almost did not change significantly (as shown in the figure). This also provides more possibilities for its controllable release as a guest molecule.
Comments (10) Figure 8 does not show the rate of coating, but are just DLS and UV traces. Please rephrase "comparativley high rate" as rate is not something that is measured in Fig 8.
Re: Thank you for your kind suggestion. We have revised the Figure 8 and corresponding text description.
Comments (11) Comments on the Quality of English Language
English can be improved, especially on page 11 bottom part (cursory research = curiosity research)
Re: I appreciate your comments on the quality of English language in my manuscript. I understand that clear and accurate language is crucial for ensuring that the research is understood. And I recognize that there are areas where I may have fallen short in expressing myself clearly or precisely in English as a non-native speaker. To address these concerns, I have carefully reviewed and revised the manuscript, focusing on improving the language quality. I am confident that these revisions will significantly improve the readability and comprehensibility of my manuscript.
Thank you once again for your attention and guidance on my manuscript. I look forward to receiving further feedback from you.
Best regards,
Author

Round 2
Reviewer 1 Report
Comments and Suggestions for Authors
The authors have addressed most of the comments and made revision of the manuscript. I thus suggest to accept it.
The writing has been improved.
Reviewer 3 Report
Comments and Suggestions for Authors
The revised version has addressed all concerns raised and is therefore fine to be published